colocr: an R package for conducting co-localization analysis on fluorescence microscopy images

Ahmed Mahmoud
Lai Trang Huyen
Kim Deok Ryong drkim@gnu.ac.kr
Department of Biochemistry and Convergence Medical Sciences and Institute of Health Sciences, Gyeongsang National University School of Medicine , JinJu , Republic of Korea
Haraguchi Tokuko
Electronic publication date: 2019 Jul 4
Publication date: 2019
Volume: 7
Electronic Location ID: e7255
Received 2019 Mar 22; Accepted 2019 Jun 5
Copyright: ©2019 Ahmed et al.
Copyright year: 2019
Copyright holder: Ahmed et al.
License: This is an open access article distributed under the terms of the Creative Commons Attribution License, which permits unrestricted use, distribution, reproduction and adaptation in any medium and for any purpose provided that it is properly attributed. For attribution, the original author(s), title, publication source (PeerJ) and either DOI or URL of the article must be cited.
License URL: https://creativecommons.org/licenses/by/4.0/

Keywords: R package, Colocalization, Image analysis, Fluorescence microscopy, Statistics

Funding: National Research Foundation of Korea (NRF) Ministry of Education Science and Technology 2018R1D1A1B07043715 Ministry of Science, ICT and Future Planning NRF2015R1A5A2008833 This study was supported by the Basic Research Program through the National Research Foundation of Korea (NRF) by the Ministry of Education Science and Technology (2018R1D1A1B07043715) and the Ministry of Science, ICT and Future Planning (NRF2015R1A5A2008833). The funders had no role in study design, data collection and analysis, decision to publish, or preparation of the manuscript.

==============================
Background

The co-localization analysis of fluorescence microscopy images is a widely used technique in biological research. It is often used to determine the co-distribution of two proteins inside the cell, suggesting that these two proteins could be functionally or physically associated. The limiting step in conducting microscopy image analysis in a graphical interface tool is the selection of the regions of interest for the co-localization of two proteins.

Implementation

This package provides a simple straightforward workflow for loading fluorescence images, choosing regions of interest and calculating co-localization measurements. Included in the package is a shiny app that can be invoked locally to interactively select the regions of interest where two proteins are co-localized.

Availability

colocr is available on the comprehensive R archive network, and the source code is available on GitHub under the GPL-3 license as part of the ROpenSci collection, https://github.com/ropensci/colocr.

Introduction

Biologists use fluorescence microscopy imaging techniques in a variety of applications. Among them, the most widely used application is the co-localization analysis. It is often used to describe the co-distribution of two proteins that are functionally linked in the cell. The underlying assumption of this technique is that two proteins closely localizing will interact with each other to potentially share some common characteristics in the cell functions. Several methods are developed for quantifying the co-localization of the two intracelullar proteins using fluorescence microscopy images. Nevertheless, co-localization analysis still has some limitations. Therefore, methods to deal with these limitations such as correcting for chromatic shift of the color components were also proposed (Manders, 1997; Manders, Verbeek & Aten, 1993; Matsuda et al., 2018).

Multiple tools implement these methods with easy to use graphical interfaces such as Fiji- an extension of ImageJ (Schindelin et al., 2012; Schneider, Rasband & Eliceiri, 2012). imager and magick are two R packages that can be used for similar image analysis (Barthelme, 2018; Ooms, 2018). Selecting the regions of interest (ROI) in a graphical interface is a critical step for these image analyses. Often, this requires manual work by the user, which can be time consuming when processing tens or hundreds of images. Also, this analysis would be very hard to reproduce or rerun with minor parameter changes. Other image analysis programmatic tools have a wider functionality and goals beyond simple analysis, so non-experienced users might have a hard time using them.

Here, we present a simple package called colocr that can be used in R environment (R Core Team, 2017). colocr enables quantifying the co-localization of two coloring dyes from the high quality microscopy images obtained from staining with two different fluorescent probes. The functions in colocr map to the intuitive steps of the co-localization analysis and do not require prior knowledge of image analysis or advanced R. The package offers a graphical user interface based on the popular Shiny applications that can be launched locally or accessed online (Chang et al., 2016).

Materials & Methods

Data sources

The confocal fluorescence microscopy images presented in this article are from the DU145 prostate cancer cell line. In this experiment, the cell line was treated with two primary antibody probes for two proteins RKIP and one of MAP1LC3B, PIK3CB, TBC1D5 or TOLLIP, and subsequently with two secondary antibody probes conjugated by different fluorescent dyes (Ahmed et al., 2018). The aim of this experiment is to determine the degree of co-localization of two proteins in this cell line and to further describe their functional association in autophagy during the tumor progression.

The DU145 human prostate cancer cells were seeded on cover glasses and cultured in DMEM containing 10% fetal bovine serum (FBS) at 37 °C in 5% CO2 humidified atmosphere. For Immunostaining, each sample was simultaneously incubated with two primary antibodies (5–20 µg/mL each) RKIP/PEBP1 (polyclonal rabbit Ab, sc-28837) and one of monoclonal mouse antibodies (LC3/MAP1LC3B, sc-376404; PIK3CB, sc-376641; TBC1D5, sc-376296; or TOLLIP, sc-136152) in 1% BSA in PBST (PBS + 0.1% Tween 20) at 4 °C overnight. Two proteins were visualized by staining with two fluorescence-conjugated secondary antibodies (anti-mouse IgGkBP-CFL 594, sc-516178, and anti-rabbit IgG Alexa FLuor 488, A27034) in PBST + 1% BSA for 60 min at 37 °C under dark. Nuclei were stained with Hoechst (300 ng/mL in 1% BSA in PBST for 10 min). All images were obtained under the confocal microscope Olympus FV 1000 (Olympus Corporation, Tokyo, Japan).

Co-localization measurements

The following is a brief discussion of the theory and interpretations of the different measurements we used in this package as measures of cellular co-localization. The articles by Manders, Verbeek & Aten (1993) and Dunn, Kamocka & McDonald (2011) describe the formal details of the statistics. For each of the co-localization measurement, we provide a definition, formula, range of values, interpretation and the suitable situations where it can be used.

Pearson’s correlation coefficient

Pearson’s correlation coefficient (PCC) is the co-variance of the pixel intensities from the two channels. The mean of the intensities is subtracted from each pixel which makes the coefficient independent of the background level. The PCC is calculated as follows:

PCC=∑iRi−R ¯×Gi−G ¯∑iRi−R ¯2×∑iGi−G ¯2

Where Ri and the Gi is the intensities of the magenta and green channels and the R ¯ and G ¯ are the average intensities. The values of PCC are between 1 and −1 for perfect correlations in the positive and negative directions respectively and 0 means no correlation. PCC measures both the occurrence and the proportionality of the pixel intensity, therefore is expected to be used in cases where the two dyes are expected to co-localize and to scale linearly.

Manders overlap coefficient

Manders overlap coefficient (MOC) is the fraction of pixels from each channel with values above the background. It doesn’t require subtraction of the mean. Therefore, the values are always between 0 and 1. The MOC is calculated as follows:

MOC=∑iRi×Gi∑iRi2×∑iGi2

Where Ri and the Gi is the intensities of the magenta and green channels. MOC is suitable to use in cases where the signal from the two proteins are expected to co-occur but not in proportion to each other.

Data objects & methods

colocr uses an S3 object called cimg from the imager package. All methods take this object as input with exception of image_load which takes a single argument for the path to the image file. image_load and roi_select returns the same cimg with an additional attribute called label in the latter case. roi_show and roi_check return NULL and four and two plots, respectively. roi_test returns a data.frame. Table 1 summarizes the input and output of each function in the package.

Table 1 Description of the package functions, inputs and outputs.

Function	Input	Output	
image_load	File path to image (string).	Image object (cimg).	
roi_select	Image object (cimg) and parameters to select regions of interest.	Image object (cimg) with and label attribute.	
roi_show	Image object (cimg) with and label attribute.	Four plots. Original image, low resolution selected regions and two gray scale images of two channels with highlighted selected regions.	
roi_check	Image object (cimg) with and label attribute.	Two plots. Scatter plot and density distribution of the pixel intensities from the selected regions in two channels.	
roi_test	Image object (cimg) with and label attribute.	A data.frame. With a column for each of the requested co-localization statistics and a row for each of the regions of interest.	

Source code & reproducibility

The source code for the package is available on GitHub under the GPL-3 license as part of the ROpenSci on-boarding repository (https://github.com/ropensci/colocr). The code and the image in this document are available at https://github.com/BCMSLab/colocrart. A simplified version of this code is presented in the last section of this article. The full version of the code is provided in an additional file.

Results & Discussion

Here, we introduce an example from the published literature where images from the DU145 prostate cancer cell line stained with dyes for two proteins RKIP/PEBP1 and LC3/MAPLC3B (Ahmed et al., 2018). The aim of this experiment is to determine how much of the two proteins are co-localized or co-distributed in the particular cell line (Fig. 1).

Figure 1 Merge image and the first and second channels on the gray scale.

Fluorescence microscopy images of (A) merged image, (B) first and (C) second channel in gray scale. DU145 cells were stained using antibodies for RKIP and LC3 as described in Data Sources. Images were obtained under the confocal microscope Olympus FV 1000. Blue represents the Hoechst-stained nuclei. Image size is 100 µm by 110 µm.

Selecting regions of interest (ROI)

The function roi_select relies on different algorithms from the imager package. However, using the functions to select the ROIs require no background knowledge in the workings of the algorithms and can be done through trying different parameters and choosing the most appropriate ones. Typically, one wants to select the regions of the image occupied by a cell or a group of cells. The package can also select certain areas/structures within the cell if they are distinct enough. The default behaviour is to select the largest contiguous region of the image and add the next (n) largest regions using the n argument.

The selection of ROIs is achieved using morphological operations from imager (Barthelme, 2018). In brief, we start by selecting the structures in the gray-scale image using the default values of three major operations; threshold, grow (dilation) and shrink (erosion). Thresholding excludes the pixels below a certain value. Grow and shrink test for whether a number of pixel outward and inward, respectively, belong to the structure. The combination of the two operations; fill and clean can include and exclude gaps in the structure, respectively. In our experience, a suitable selection can emerge easily by varying these parameters in a trial and error fashion.

This function returns a cimg object containing the original input image and an added attribute called label to indicate the selected regions. label is a vector of integers; with 0 for the non-selected areas, 1 for the first, 2 for the second selected regions and so on. The selection process can be assessed visually using roi_show. The function outputs four plots; the merge image, the pixel set and each of the two channels with highlighted ROIs (Fig. 2).

Figure 2 Selection of regions of interest.

(A) Merge image is the overlap of magenta, green and blue dyes. (B) Pixel set is a low-resolution image of the selected regions of interest. (C & D) Channel one and channel two with highlighted regions of interest (red line). Image size is 100 µm by 110 µm.

Quality assessment of pixel intensities

Both the co-localization measurements implemented in this package quantify different aspects of the linear trend between the pixel intensities from the two channels of the image. Therefore, it is useful to visualize this trend and the distribution of the intensities to make sure whether the analysis is suitable. The expectation is that the pixel intensities from the two channels should align with the diagonal in the first graph and show nearly overlapped distributions in the second with the similar pattern of pixel values (Fig. 3).

Figure 3 Scatter and density distribution of pixel intensities.

(A) The raw pixel intensities of the channels one and two from the three regions of interest (colors) in Fig. 2 are shown as points. (B) The density of the pixel values of the first (magenta) and second (green) channels from all regions are shown as lines.

Calculating co-localization measurements

The two different measurements implemented in this package are the PCC and MOC. We described the rational and the formulation of those measurements in “Materials & Methods”. Invoking the test is a one function call on the selected regions of interest. roi_test returns a data.frame with a column for each of the desired measurements and a row for each of the selected regions (n) (Table 2).

Table 2 Co-localization statistics.

ROI	PCC	MOC	
1	0.87	0.94	
2	0.88	0.93	
3	0.85	0.94	
Average	0.87	0.94	

Testing for statistical significance

While colocr doesn’t implement any formal statistical tests for significance, it is an important issue to discuss. One can test the significance of the difference in co-localization between two groups (co-localized vs uncorrelated probes) using a simple t-test. Alternatively, one can compare the observed co-localization measurement in one group to a null model generated from the same data. Dunn, Kamocka & McDonald (2011) discussed the difficulties in generating true random models to compare with the observations. For the purposes of comparison, probes that don’t co-localize with the protein of interest (negative control) can be used. This comparison can be tested using a t-test when the observations are normally distributed, otherwise non-parametric tests can be used.

Other features

Processing a collection of images

To process a collection of images at once, the input for the functions should be lists of the original object type. Other parameter arguments can be single values that apply to all images or as lists of the same length with specific values for each image. Similarly, the output of image_load, roi_select and roi_test would be a list of the original output object type. For roi_show and roi_check, the output is the same set of plots for each image.

Graphic user interface (Shiny application)

Arguably, selecting the regions of interest is the most time-consuming step in this analysis. Usually, one has to select the regions by hand when using image analysis software such as ImageJ. This package only semi-automates this step, but still relies on the user’s judgment on which parameters to use and whether the selected ROIs are appropriate. To simplify this step, the package provides a simple shiny app to learn these parameters interactively and use it in the rest of the workflow. This app can be invoked locally from within an R session or accessed online at the following address: https://mahshaaban.shinyapps.io/colocr_app2/.

Other image processing packages in R

The three main image processing packages available in R are imager, magick and EBImage (Barthelme, 2018; Ooms, 2018; Pau et al., 2010). imager wraps the CImg and magick wraps the Magick++ C++ libraries, respectively (Tschumperle, 2018; Bob Friesenhahn, 2018). Both packages and their underlying libraries contain a wide functionality for image processing and analysis. colocr uses some imager and magick functionality to simplify the co-localization analysis of microscopy images. Similarly, EBImage can be used to select areas of interest in images and extract pixel intensities. In colocr, there are only a few high-level functions that map directly to the steps of the co-localization analysis. The users don’t have to worry about much of the details of the data structures or the specifics of the applied morphological operations. Finally, the functions in this package are vectorized and can be used to process multiple images at once. The current implementation cannot handle 3D or time-series images. In addition, only common bitmap and raster image types are supported by the read function. Together, colocr uses existing image processing R packages to create a custom tool specific for the co-localization analysis.

A case study from the published literature

We used the colocr package to reproduce an analysis from the published literature for the co-localization of RKIP with four different proteins in DU145 cell line (Ahmed et al., 2018). The amount of co-localization of RKIP was quantified with each of the proteins in more than 5 images each and represented as PCC and MOC values (Fig. 4). The quantification was originally conducted using the ImageJ Fiji plugin and was found to agree with colocr calculations in both PCC and MOC values, suggesting that our co-localization R package is very compatible with the ImageJ Fiji plugin.

Figure 4 Co-localization of RKIP/PEBP1 with autophagy-related gene products.

(A) Co-localization images between RKIP/PEBP1 and autophagy gene products (MAP1LC3B, PIK3CB, TBC1D5 and TOLLIP) in a human prostate cancer cell line DU145. Scale 10 µm. (B) The degree of co-localization measured as Pearson’s correlation coefficient (PCC) and Manders overlap coefficient (MOC) (n > 15) using both colocr and ImageJ. Numbers inside the graph indicate average coefficient values for each co-localization.

Typical colocr workflow

A typical colocr workflow starts by loading the merge images in an R session using image_load. Then selecting the regions of interest using roi_select. Finally, calculating the desired co-localization measurement using roi_test. Optionally, roi_show highlights the selected regions on the images and roi_check visualizes the scatter and the density distributions of the pixel intensities. Figure 5 depicts the steps and the functions of the typical workflow.

Figure 5 Work flow of the co-localization analysis in the colocr package.

The diagram depicts a typical workflow for using colocr. This includes loading the merged image, selecting the regions, extracting the pixel intensities and calculating the co-localization measurement.The labels in blue are the specific functions in colocr to perform each step of the workflow.

Reproducing figures and table in this document

In this section, we simplified a version of the code used to produce this document. Briefly, we load the required R libraries, construct a path to the image file (example image) and apply a typical workflow to calculate the co-localization measurements.

First, we start by loading the two libraries imager and colocr.

library(imager) library(colocr)

The example image used throughout the document is from DU145 cell line stained for RKIP and LC3 in the first and second channel, respectively. The image is included in the package and can be accessed using system.file.

# get  image  path fl <− system.file('extdata', 'Image0003_.jpg', package = 'colocr')

We load the image using image_load and show it along with the two channels (Fig. 1).

# load  images  and  channels img <− image_load(fl) img1 <− channel(img, 1) img2 <− channel(img, 2)

# generate  figure  of  images  and  channels par(mfrow = c(1,3), mar = c(0, 0, 1, 0)) plot(img, axes = FALSE, main = 'Merge') plot(img1, axes = FALSE, main = 'Channel One') plot(img2, axes = FALSE, main = 'Channel Two')

Typically, one would use the roi_select to choose the regions of interest as a first step in the analysis workflow. roi_show highlights the selected regions (Fig. 2).

# select  regions  of  interest par(mfrow = c(2,2), mar = c(0, 0, 1, 0)) img %>%    roi_select(threshold = 90,                 shrink = 10,                 fill = 5,                 clean = 10,                 n = 3) %>%    roi_show()

Next, roi_check shows the scatter and the density distribution of pixel intensities from the selected regions of interest (Fig. 3).

# check  pixel  intensities par(mfrow = c(1,2), mar = c(4, 4, 1, 1)) img %>%    roi_select(threshold = 90,                 shrink = 10,                 fill = 5,                 clean = 10,                 n = 3) %>%    roi_check()

Finally, roi_test calculates the co-localization measurements (Table 2).

# calculate co−localization  stats img %>%    roi_select(threshold = 90,                 shrink = 10,                 fill = 5,                 clean = 10,                 n = 3) %>%    roi_test(type = 'both')

Conclusion

colocr implements a simple workflow for the co-localization analysis of fluorescence microscopy images. The package provide functions for selecting regions of interest, extracting the pixel intensities and calculating the co-localization measurements.

Supplemental Information

Supplemental Information 1 R code to obtain the example images and reproduce the figures and tables

Click here for additional data file.

We thank all lab members for the critical discussion during developing this R package. We thank the ROpenSci team for reviewing the source code and hosting the package on their repository.

Additional Information and Declarations

Competing Interests

Author Contributions

Data Availability

The authors declare there are no competing interests.

Mahmoud Ahmed conceived and designed the experiments, analyzed the data, prepared figures and/or tables, authored or reviewed drafts of the paper, approved the final draft.

Trang Huyen Lai performed the experiments, analyzed the data, approved the final draft.

Deok Ryong Kim conceived and designed the experiments, contributed reagents/materials/analysis tools, authored or reviewed drafts of the paper, approved the final draft.

The following information was supplied regarding data availability:

The source code of the R package is available at https://github.com/ropensci/colocr. The images in the article are available at https://github.com/BCMSLab/colocrart.

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
