# Peer review of "colocr: an R package for conducting co-localization analysis on fluorescence microscopy images"

_PeerJ, doi:10.7717/peerj.7255_

## Round 0.1 · original submission · Major Revisions

I think the reviewers comments are constructive and helpful to improve your manuscript. I hope you revise the manuscript accordingly.

# Reviewer 1 ·

Basic reporting

1. The manuscript by Ahmed et al. reports their software for calculating two major colocalization coefficients, i.e. Pearson’s Correlation Coeffient (PCC) and Mandars Overlap Coefficeint (MOC), in multicolor imaging of biological samples. The manuscript is readable and english grammar is mostly fine. However, there are a few paragraphs with unclear intensions or discussions at inappropriate places. The manuscript is inclined too much toward software description, but lacks insights into the practical issues or theories in microscopy. For these reasons, this reviewer requests a major revision before publication from PeerJ.
2. The manuscript did not mention the practical problems in multicolor microscopy and thus failed to add key references in this field. Firstly, the authors should mention in their Introduction or Discussion section about chromatic shift correction, which is an important pretreatment before measuring colocalization of microscopy data. The milestone papers to cite are:
(i) Manders, E. M. M. Chromatic shift in multicolour confocal microscopy. J. Microsc. 185, 321–328 (1997).
(ii) Churchman, L. S., Ökten, Z., Rock, R. S., Dawson, J. F. & Spudich, J. A. Single molecule high-resolution colocalization of Cy3 and Cy5 attached to macromolecules measures intramolecular distances through time. Proc. Natl. Acad. Sci. U. S. A. 102, 1419–1423 (2005).
(iii) Matsuda, A., Schermelleh, L., Hirano, Y., Haraguchi, T., and Hiraoka, Y. Accurate and fiducial-marker-free correction for three-dimensional chromatic shift in biological fluorescence microscopy. Sci. Rep. 8: 7583 (2018).
3. Secondly, the authors should cite the original theoretical paper describing application of PCC and MOC in biological microscopy.
(i) Manders, E. M. M., Verbeek, F. J. & Aten, J. A. Measurement of co‐localization of objects in dual‐colour confocal images. J. Microsc. 169, 375–382 (1993).
4. The manuscript may require additional data sets for testing the validity of the approach. Detailed suggestions will be made in “Validity of the findings”.
5. Presentations of tables and figures need to be re-considered. Detailed suggestions will be made in the “General comments” section.

Experimental design

This work reports software development using R programming language. Their software helps to measure the amount of colocalization, that is widely required in biological microscopy. The software is open-source and its quality is validated by the ROpenSci community. This work fits with aims and scope of PeerJ.

Validity of the findings

1. The difficult step toward quantifying colocalization is the determination of ROI. While authors developed a software facility to select ROI, the image presented in the manuscript is only a 2D image. Does it work for 3D images? In the case of 3D images of wide-field microscopy or confocal microscopy using large pinhole sizes, out-of-focus blur makes simple thresholding difficult. The authors should test if their approach works for 3D images, or if it doesn’t then mention its limitation about the dimension.
2. ‘Additional file 1’ is mentioned at the end of “Materials and Methods”, but it was not found; only the manuscript was found.

Additional comments

1. Specific names of packages or functions (Such as ‘Shiny’, ‘cimg’, ‘imager’, ’image_load’, ‘data.frame’) should be in different fonts, capitalized if applicable, or enclosed by quotations to discriminate from conventional words.
2. What image file formats are accepted by the software? Are there any restrictions for image data (data size, bit depth, number of channels, number of time frames etc..)?
3. The word “statistics” is used throughout to indicate ‘statistics’ or ‘coefficient’. When talking “colocalization statistics”, biologists would expect a statistical analysis of the amount of colocalization measured for many cells. Therefore, I would recommend to use ‘colocalization’, ‘measurement of colocaliztion’ or ‘quantifying colocalization’ for a single measurement, and ‘statistics’ for many cells. In this context, ‘statistics’ seems inappropriate, for example, in line 62 and probably line 130, and many other places if they are intended to mean ‘quantifying colocalization’.
4. Line 82, “Testing for statistical significance” this subsection is a discussion which is inappropriate for the “Materials and Methods” section. Furthermore, the conclusion of this subsection is unclear. Please explain the last sentence more carefully. Why “auto-correlation in the image pixels” in the null model is a problem? What is “the varying nature of the probe interaction”? Do the authors finally recommend to use t-test?
5. Line 73, this sentence needs proof-reading.
6. Line 89, Table 1 and Line 164. My personal impression is that ‘how to use’ of the code or too much detail of individual functions should be written in the document of the source code and unnecessary in the main body of a scientific manuscript. Alternatively, can these be moved to supplemental files? I think Figure 4 is sufficient to introduce their functions.
7. Line 102, “cancer cell line were stained…” should be changed to “cancer cell line stained…”
8. Figure 1-2. Is the blue color in the merged image really required? Because of this blue color, it is very difficult to see the red color. The images are too small to validate that the colocalization coefficients are really true at the pixel level. A magnified view would be helpful. Also since the combination of red and green may be difficult to see for those with color-blindness, a combination of magenta and green or red and cyan is recommended. Scale bars are missing.
9. Line 109, “However” is inappropriate here. Just removing it makes the sentence more readable.
10. Figure 3. The color code in the left plot is ambiguous. It is difficult to see how many colors are used in the plot, therefore, it is unknown how many ROIs are used here. Do these ROIs correspond to ROIs in Figure 2 and Table 2? If so, please indicate as such. Also in this figure legend, at the end of the first sentence, ‘show’ should be changed to ‘shown’.
11. Line 131-132, ‘We described … elsewhere.’ Please indicate where it is described.
12. Line 156, the message in this paragraph is unclear. Does the author describing or discussing about the “other image processing packages in R”? The last sentence needs proof-reading (where is verb?).

·

Basic reporting

This manuscript reports a new package for R for co-localisation analysis of fluorescent microscopy images called “colocr”. The package is open source, and is available on GitHib but there is no licence on the software. In this report the authors present the use of the software against one pair of individual images which seems insufficient to evaluate the utility of the software.

Experimental design

No comment

Validity of the findings

The authors compare their work on colocr to a few co-localisation packages that are part of the ImageJ plugin collection. There are several other Image J plugins and indeed several R packages already released that implement colocalization. For example, the Bioconductor package and in particular EBimage R suite contains many of the functions the authors report here. In addition, the arrival and universal adoption of super-resolution methods has raised several questions regarding the utility of colocalization at the diffraction limit, since super-resolution techniques will often resolve what is co-localised in conventional microscopy. Thus while the work appears to be well formed the rationale for yet another co-localisation package hasn’t been well presented.

Additional comments

If the author’s present a comprehensive representation of tools for co-localisation in ImageJ and R and elsewhere, a clear argument for their use and the description of colocr, the paper would then be suitable for publication.

---

## Round 0.2 · Minor Revisions

One of the reviewer raises a few questions. I think the comments are useful for improve the manuscript. Please revise the manuscript according to the comments.

Reviewer 1 ·

Basic reporting

The manuscript was improved after the revision. There still remain a few points that are confusing for readers. This reviewer requests a minor revision for this manuscript.

Experimental design

No comment.

Validity of the findings

No comment.

Additional comments

At line 30, the references seem to be cited for a wrong reason. “Churchman et al., 2005” and “Matsuda et al., 2018” solved the problem of chromatic shift and therefore should be cited at line 32 together with “Manders 1997”.

Related to the citations, there are several references missing volume and page numbers in the REFERENCES section.

We can predict that there are many researchers who wish to measure colocalization of 3D images or time series. To avoid these researchers spending time for nothing and to help them re-direct to other resources, the authors should indicate the dimensions that their current software version can handle in the manuscript. Also, even though the program uses “magick” package to read image files, many file formats of microscope companies (such as czi, oib, lif etc..) does not seem to be readable. Again to save time for researchers not spending time to try directly load their image files from microscopes (which is possible in the case of, for example, ImageJ), please indicate the kind of files that are accepted by colocr. Such descriptions should help readers to make a fair judgement on software and be consistent with the idea to make the manuscript self-contained as the authors claimed in their response.

Line 78-79 needs proof reading (spelling of “therefore” and overlapping “to”s).

Figure 2 and 3. The authors can indicate the size of image in the legend instead of inserting scale bars.

---

## Round 0.3 · accepted · Accept

I am happy to accept this manuscript for publication in PeerJ.